# Myocardial Injury after Stroke

**DOI:** 10.3390/jcm11010002

**Published:** 2021-12-21

**Authors:** Michal Mihalovic, Petr Tousek

**Affiliations:** Department of Cardiology, Third Faculty of Medicine, University Hospital Královské Vinohrady, Charles University, 100 34 Prague, Czech Republic; petr.tousek@fnkv.cz

**Keywords:** stroke, autonomic dysfunction, troponin, arrhythmias, cardiac injury

## Abstract

The cardiovascular system is markedly affected by stress after stroke. There is a complex interaction between the brain and heart, and the understanding of the mutual effects has increased in recent decades. Stroke is accompanied by pathological disturbances leading to autonomic dysfunction and systemic inflammation, which leads to changes in cardiomyocyte metabolism. Cardiac injury after stroke may lead to serious complications and long-term cardiac problems. Evidence suggests that blood biomarkers and electrocardiogram analyses can be valuable for estimating the severity, prognosis, and therapy strategy in patients after stroke. It is necessary to distinguish whether these abnormalities presenting in stroke patients are caused by coexisting ischemic heart disease or are caused by brain injury directly. Distinguishing the origin can have a great impact on the treatment of patients after acute stroke. In this article, we focus on epidemiology, pathophysiological mechanisms, and the presentation of cardiac changes in patients after stroke.

## 1. Introduction

Stroke is the second most common cause of death and the main cause of disability worldwide. It is a major public health problem and is associated with a significant socioeconomic burden, because patients often have functional impairment and disabilities after a stroke [1]. After acute cerebral injury, patients are vulnerable to cardiac complications and frequently show signs of myocardial injury. Evidence implies that cardiac complications and mortality after acute stroke are not merely caused by concomitant coronary artery disease, but also through direct interaction between the brain and heart [2]. Neurological disorders affecting the cardiovascular system are defined as a disruption of the heart–brain axis. Neuro-cardiology is a specialty that studies the interaction between these two systems. Sympathetic hyperactivity, the hypothalamic–pituitary–adrenal axis, immune and inflammatory responses, and gut dysbiosis are pathophysiological mechanisms involved in dysregulation after acute stroke [3,4,5]. Little is known about the clinical consequences of electrocardiographic (ECG) and laboratory changes.

## 2. History

The brain–heart connection was first described in the early 20th century. In 1901, Cushing described how increased intracranial pressure was often present with increased blood pressure and a slower heart rate. In 1914, Levy in his work showed that changes in the central nervous system (CNS) can lead to changes in cardiac function and cause arrhythmias [6]. A later study from Beattie et al. showed that ventricular premature systoles can be produced by stimulating the hypothalamus, and in 1954, Burch et al. first described the specific ECG patterns in 17 patients after intracranial hemorrhage, observing frequent QT interval prolongation, U waves, and large amplitude T wave [7,8]. Subsequent studies have examined arrhythmias, repolarization and conduction abnormalities in patients after acute stroke, including subarachnoid hemorrhages (SAH), intracerebral hemorrhages (ICH), acute ischemic stroke (AIS), and transient ischemic attacks (TIA) [9,10,11,12].

## 3. Epidemiology

The Framingham study showed that the incidence of stroke doubled in the presence of coronary heart disease, tripled with arterial hypertension, increased four-fold with cardiac failure, and increase five-fold with atrial fibrillation [13]. Cardiovascular complications are among the leading causes of mortality after acute stroke. Cardiac events after acute stroke include myocardial infarction, heart failure, abnormal heart rhythms, and in some cases cardiac arrest. The incidence of cardiovascular events or pathological cardiac findings after acute ischemic stroke ranges from 3% for myocardial infarction to >50% for new ECG changes [14,15]. Moreover, cardiac causes are responsible for 2–6% of the mortality 3 months after an AIS, and approximately 19% of patients have a fatal or a major non-fatal cardiac event during this period [16]. The most serious complications after an AIS are in the acute phase. Furthermore, impaired cardiac function after an AIS increases the risk of worse neurological outcomes and 90-day disability [17,18]. Patients after SAH develop cardiac arrhythmias in 5% of the cases, and 80% of the patients develop ECG changes within the first year. This is also associated with a poorer outcome [11,19].

## 4. Pathophysiology

The pathophysiological mechanisms by which stroke leads to ECG abnormalities and myocardial injury are not fully understood. Leading mechanisms involve the hypothalamic–pituitary–adrenal axis (HPA), immune response and other factors such as inflammation and gut dysbiosis. Disturbances in these systems lead to changes in cardiomyocyte metabolism [2,20,21].

### 4.1. Autonomic Dysfunction

One of the leading theories explaining myocardial injury following stroke involves altered autonomic balance. The central nervous system regulates autonomic responses from the brain to the heart; these are involved in both physiological and pathological responses [22]. Changes in the central structures have direct effects on the autonomic nervous system (ANS) and may lead to excessive sympathetic stimuli after acute stroke, the connection is provided by sympathetic pre-and post-ganglionic neurons. This leads to the activation of β-receptors and the consequent activation of cyclic adenosine monophosphate-protein kinase A signaling, resulting in the release of intracellular calcium from the sarcoplasmic reticulum. The abnormal release of the calcium into the cells causes contractile dysfunction and adenosine triphosphate (ATP) depletion, which causes mitochondrial dysfunction and cardiac myocyte damage that can be reversible or result in cell death [3]. Central structures controlling parasympathetic function include medulla oblongata, nucleus ambiguous, reticular formation, and the nucleus of nervus vagus functioning through the epicardial ganglion plexuses and postganglionic nerve fibers, releasing acetylcholine. Parasympathetic activity is provided by the muscarinic receptor which reduces cyclic adenosine monophosphate, which can lead to a reduction in contractility by slowing the depolarization. The HPA axis includes the hypothalamus, pituitary, and adrenal glands. After an acute stroke, adrenal glands release cortisol and through catecholamines activate β1 adrenoreceptor leading to excessive calcium release, depletion of ATP, and oxidative stress, therefore suggesting that catecholamine released in the circulation of myocardial nerve endings can cause cardiac toxicity [22] (Figure 1).

### 4.2. Immune Response

Cerebral hemorrhages and ischemic strokes often lead to a systematic inflammatory response [23]. Stroke damages plasma membranes, resulting in an increase in ATP with further activation of microglial cells and production of inflammatory cytokines such as IL-2, IL-6, myeloperoxidase, and integrins, which stimulate oxidative stress [4,24]. Inflammatory cytokines accumulate on endothelial cells, destroying the collagen in the atherosclerotic plaque. The weakened fibrous envelopes can lead to coronary events [25].

### 4.3. Other Contributing Factors

Other contributing mechanisms include interactions among the intestinal flora, CNS, and cardiovascular system. Some studies have suggested that patients suffer from intestinal permeability disruption after acute stroke [26]. After a stroke, altered intestinal permeability results in the translocation of bacteria and endotoxins to the bloodstream. This increases pro-inflammatory cytokines and systemic inflammation that can aggravate myocardial damage [5]. Moreover, the translocation of bacteria and endotoxins influences blood metabolites, indoxyl sulphate and trimethylamine-N-oxide. Indoxyl sulphate affects cardiac remodeling via NFkB [27]. Trimethylamine-N-oxide is associated with cardiac dysfunction, heart failure, and enhancing thrombosis [28].

## 5. Does Infarction Site Matter?

There is considerable evidence of the important role of brain areas in modulating cardiac function. Cerebral structures are involved in cardiac activity mainly via the sympathetic and parasympathetic nervous systems, which modify the metabolic balance, cardiomyocyte contraction, and the heart rate [29]. Some studies have shown right-sided dominance in the brain for sympathetic cardiovascular effects [30,31]. There has been a focus on the involvement of the insula in regulating cardiac functions [32]. The insula cortex affects several autonomic responses. It projects directly to the lateral hypothalamus, parabrachial nucleus, and nucleus of the solitary tract, which in turn projects directly to sympathetic pre-ganglionic areas. Studies have reported an association between the insula cortex and cardiac events. There is also evidence of insular lateralization, in which damage to the right insula cortex may increase sympathetic activity and cause cardiac injury, while left insular cortex damage may lead to increased parasympathetic activity [33]. Cechetto et al. demonstrated increases or decreases in arterial blood pressure accompanied by changes in sympathetic nerve activity induced by specific insula areas [34]. In a study of 62 patients, Tokgozoglu et al. showed that stroke in the right insula led to decreased heart rate variability and increased the incidence of sudden death, which suggests possible sympathetic dominance after stroke, which could increase the incidence of ventricular arrhythmias [35]. Several studies have shown that after a stroke involving the insula, patients have ECG abnormalities and increased norepinephrine, brain natriuretic peptide (NT-proBNP), and troponin (cTn) levels [10,36,37].

The importance of the insula in controlling the autonomic system and its regulation of the cardiovascular system has been supported by stimulation experiments and human stimulation studies. Oppenheimer et al. showed that phasic micro-stimulation of the rat insula cortex leads to tachycardia or bradycardia. Prolonged stimulation causes progressive heart block, increases norepinephrine levels, and causes death in the asystole. Stimulating the right anterior insular cortex in humans leads to increased blood pressure and heart rate, while stimulating the left insula results in bradycardia [38,39]. Moreover, Chouchou et al., in their study of 47 epileptic patients after insular electrical stimulation, showed that almost half had cardiac dysregulation; stimulation of the posterior insula resulted in tachycardia, whereas anterior resulted in bradycardia [40]. However, only a few studies have focused on isolated insular changes and further studies are needed to understand insular lateralization and its pathophysiological mechanisms.

## 6. Clinical Presentation of Cardiac Injury

### 6.1. Electrocardiographic Changes and Cardiac Arrhythmias

Electrocardiographic abnormalities and arrhythmias frequently follow acute stroke and can be seen both in ischemic and hemorrhagic stroke.

#### 6.1.1. Electrocardiographic Changes and Arrhythmias in Acute Ischemic Stroke

Electrocardiographic changes are seen in 50–80% of patients after AIS. The most frequent changes include a prolonged QT interval (20~35%), T wave inversion (15~35%) and ST depression (25~33%). Some of the common arrhythmias include atrial fibrillation (15–38%), ectopic beats (30%), sinus tachycardia (24%), and atrioventricular block (21%) [10,41] (Table 1). Atrial fibrillation is a risk factor for secondary complications, such as ventricular tachycardia, heart failure, or cardiac death. Atrial fibrillation, A-V block, ST-depression, ST-elevation, and inverted T waves are independent risk predictors of outcome 3 months after stroke, as assessed by the modified Rankin Scale (mRS) [10,42]. Stead et al. showed that the QTc interval was prolonged in 36% of 345 patients with AIS at admission [12]. A prolonged QTc interval was significantly associated with decreased survival after 3 months and worse neurological outcomes [12]. Increased norepinephrine levels are significantly related to QTc prolongation [43]. Moreover, in a study on 625 patients after ischemic stroke, an upright T in aVR, seen in 32.2% of patients at admission, was a significant independent predictor of death or ischemic stroke recurrence in patients after ischemic stroke [44]. ST depression and Q waves are significantly associated with increased troponin T levels [45].

#### 6.1.2. ECG Changes and Arrhythmias in Intracerebral and Subarachnoid Hemorrhage

Evidence suggests that ECG abnormalities are more common after SAH than AIS. The most common abnormalities are QT prolongation (42%), ST segment changes (37%), prominent U waves (16%), and T wave abnormalities (12%) [46] (Table 1). ST elevation, T wave inversion, and U waves are independent risk predictors of outcome after 3 months [47]. Cardiac arrhythmias occur in almost 5% of patients after SAH. The most common arrhythmias include atrial fibrillation and flutter (76%), ventricular arrhythmias (16%), and junctional rhythm (16%) [11]. Ventricular arrhythmias after aneurysmal SAH are associated with reduced survival rates after 3 months. Moreover, QTc prolongation and decreased heart rate are risk factors for developing ventricular arrhythmias [48].

There are limited data on ECG abnormalities after ICH, but in one study, they were detected in more than half of the patients. The most frequent ECG abnormalities in that study included ST depression (24%), QTc prolongation (19%), and T wave inversion (19%). All three were associated with a deep hematoma. The most frequent arrhythmias included sinus tachycardia (19.4%) and ectopic beats > 10% (25.8%). No ECG abnormalities were significantly related to neurological deficit [49]. In Van Bree et al. [50], more than 80% of patients after ICH presented with at least one ECG abnormality, most commonly QTc prolongation (36%), which was associated with insular cortex involvement and presence of interventricular blood and hydrocephalus.

### 6.2. N-Terminal Prohormone of Brain Natriuretic Peptide (NT-proBNP)

Several studies have reported elevated NT-proBNP following SAH and AIS [10,36,51,52]. In some studies, NT-proBNP was elevated in almost two-thirds of patients after AIS, peaking the day after symptom occurrence and declining thereafter [53,54]. Elevated plasma NT-proBNP levels are independently associated with stroke severity, poor functional outcome, and mortality after AIS [37,55]. Montaner et al. showed that B-type natriuretic peptide (BNP) levels are independent predictors of early mortality and neurological worsening after an acute stroke, with no difference between ischemic and hemorrhagic strokes [52]. Plasma levels of BNP are independently associated with long-term mortality [56]. In the study of almost 600 patients after ischemic stroke, measuring the BNP levels predicted mortality in patients with cardioembolic stroke [57]. NT-proBNP is correlated with the NIHSS score and stroke severity and also positively correlated with infarction size and with the modified Rankin Scale (mRS) [36,52,58].

BNP is elevated in about 75% of cardioembolic strokes. In many cases, the etiology of cardioembolic stroke is embolization from the left atrium, mostly from thrombi that form in atrial fibrillation. The presence of atrial fibrillation is associated with elevated BNP levels, and the BNP level is an independent risk factor for cardioembolic stroke [58,59]. Another study found no significant association between NT-proBNP and infarction size or stroke severity [60].

### 6.3. Troponin (cTn)

Troponins are very sensitive, specific biomarkers for the detection of myocardial injury. Levels of cTn may be increased in many other conditions, such as acute myocarditis, sepsis, pulmonary embolism, heart failure, and renal insufficiency [61].

In one study, cTn was elevated in 5–8% of patients with acute ischemic stroke [62]. Fure et al., in a study of 279 patients after AIS, showed that ST depression was significantly associated with an increase in highly sensitive troponin T (hs-cTnT) [45]. In that study, an increased hs-cTnT was also associated with a poor short-term outcome [45]. Moreover, cTn positivity at admission is an independent predictor of AIS [63].

Cardiac enzymes are elevated in 10~28% of patients after SAH [64,65]. This is associated with increased stroke severity, mortality, and worse neurological functional outcome [66]. A retrospective study of 617 patients with SAH also found increased mortality in patients with high hs-cTnI levels [67].

## 7. Detecting and Managing Myocardial Changes

Acute stroke has been known to cause cardiac abnormalities, including arrhythmias, ventricular dysfunction, myocardial infarction, or sudden cardiac death [68]. Depending on the pathophysiological mechanism, stroke can lead to neurogenic cardiac injury caused by a disruption in the ANS and catecholamine release, such as neurogenic stunned myocardium (NSM) and Tako-Tsubo syndrome (TTS), or can be caused by myocardial infarction. NSM and TTS can both present with similar changes in the ECG, alteration in cardiac function, and elevated biomarkers as myocardial infarction [69,70]. Differences between myocardial injury caused solely by neurologic alteration, myocardial infarction or isolated ECG abnormalities are sometimes impossible to distinguish. Therefore, it is essential for neurologists to examine the history of ischemic heart disease and chest pain and be careful when a patient has sudden circulatory deterioration. Other factors indicative of myocardial infarction include dynamic changes in the ST segments, new Q waves, new bundle branch block or malignant ventricular arrhythmia. Moreover, the elevation and subsequent fall in troponin suggests myocardial infarction [71,72]. Coronary angiography is indicated in patients after acute stroke only when the diagnosis of acute myocardial infarction is almost certain and when it endangers the patient more than the ongoing stroke. It is also necessary to consider administering heparin during the procedure, as it can lead to hemorrhagic transformation.

Even though many patients after stroke do not suffer from myocardial infarction, new ECG changes and elevation in biomarkers might suggest myocardial injury and higher cardiovascular risk. It is necessary that these patients are not overlooked, and they undergo detailed cardiac examination, including early echocardiographic examination to exclude wall motion abnormalities. Released patients with abnormalities presented during hospitalization should be considered for long-term cardiology dispensarization, ambulatory ECG monitoring, and control echocardiography. Moreover, high-risk patients with biomarker, ECG and echocardiographic changes should be considered for CT coronary angiogram, cardiac magnetic resonance imaging, or coronary angiography to exclude ischemic heart disease (Figure 2).

## 8. Discussion and Conclusions

There is a high prevalence of myocardial injury in patients after stroke. Cardiac injury after stroke may lead to serious complications, including long-term cardiac problems such as heart failure, or it can cause only mild recoverable damage. Blood biomarkers and ECG analyses are valuable for assessing the severity, prognosis, and therapy strategy for patients after stroke and have a great effect on clinical practice and treatment. It is important to know if these abnormalities in stroke patients are caused by coexisting heart disease or are caused by the acute brain injury directly. Therefore, a detailed cardiac examination and testing for cardiac injury are crucial for detecting cardiovascular disease. Cardiac monitoring should be considered during the acute hospitalization phase at a minimum. Patients should be followed carefully, as the most common cause of death following acute stroke is cardiovascular. Moreover, some laboratory markers, such as NT-proBNP provide useful short- and long-term prognostic information.

Many previous studies have been limited by small sample sizes. In some studies, patients had heart failure and renal insufficiency, which may affect troponin levels. Moreover, in most studies, transthoracic echocardiography was not performed. Prospective studies with defined exclusion criteria and sufficient statistical power are needed to elucidate the association between cardiac injury and clinical outcomes among acute stroke patients. 

There is evidence of a close relationship between the heart and brain. After acute stroke, many patients have cardiac complications, which may worsen their outcomes. Therefore, close cooperation between neurologists and cardiologists is needed when treating patients after acute stroke.

## Figures and Tables

**Figure 1 jcm-11-00002-f001:**
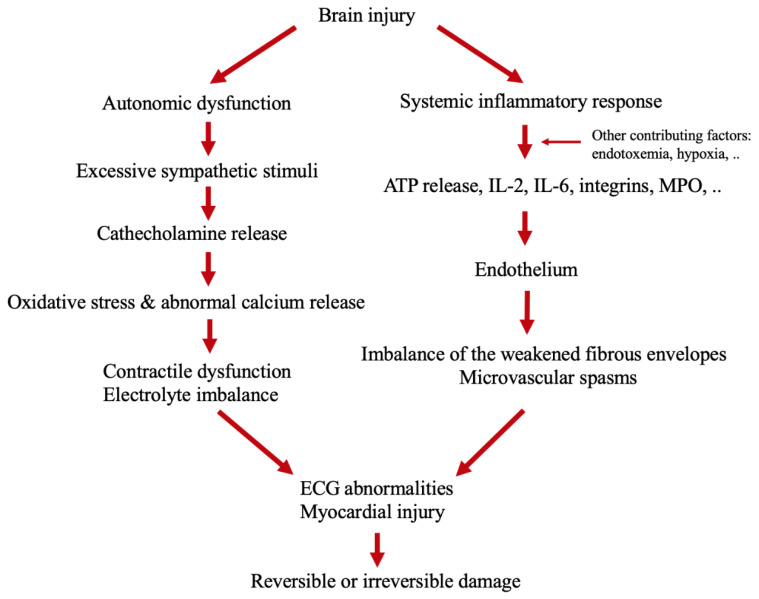
Mechanisms of the cardiac changes after brain injury [22,23,24,25,26,27,28].

**Figure 2 jcm-11-00002-f002:**
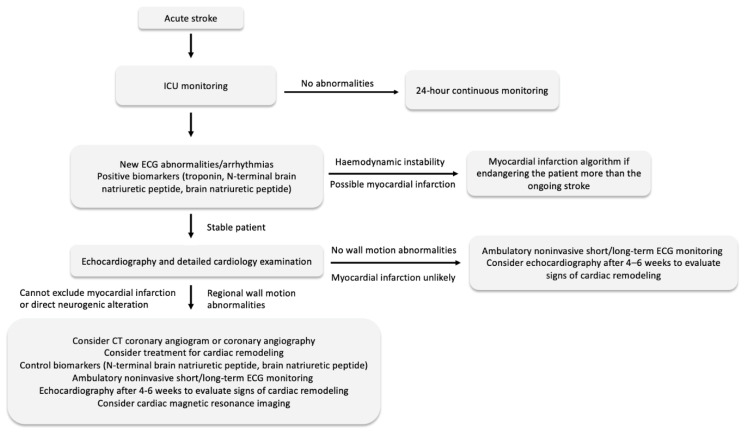
Steps for detecting and managing cardiac complications after acute stroke [69,70,71,72].

**Table 1 jcm-11-00002-t001:** Most common ECG abnormalities and arrhythmias in acute stroke patients [42,43,44,45,46,47,48].

Morphologic Changes	Conduction and Rhythm Changes
QTc prolongation	Sinus tachycardia
T wave inversion	Atrial fibrillation and atrial flutter
ST segment depression	Atrioventricular block
Prominent U wave	Atrial and ventricular ectopic beats
Unspecified ST changes	Sinus bradycardia

## Data Availability

All data used are included in this published article/below references.

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
