# Peer review of "Myocardial Injury after Stroke"

_jcm, 2021, doi:10.3390/jcm11010002_

Round 1

Reviewer 1 Report

I appreciate the opportunity to review this manuscript, which addresses a scarcely studied topic in stroke. The text is well written and reads well. The subtopics are appropriate and supported with current publications.
My only criticism is that the sixth section on detecting and managing myocardial changes is relatively short. Although the authors provide a nice figure with a proposed algorithm, the text lacks length.
Nevertheless, overall my impression is excellent, and I congratulate the authors for their effort. 

Author Response

Dear Reviewer, 

thank you very much, we highly appreciate your feedback.

As you suggested we extended sixth section with further details and proposed managements.

Sincerely, 

Mihalovic M

Reviewer 2 Report

I read this interesting manuscript on myocardial involvement in the setting of stroke. The manuscript is well-written and easy to read, providing a broad overview on a wide topic. A couple of issue should be addressed, possibly requiring independent paragraphs.

 - The heterogeneous clinical manifestation of stroke related myocardial injury should be addressed (e.g. takotsbo, isolated ECG abnormalities, myocardial stunning etc.)

 - It could be of interest if the authors could provide a more detailed description of the brain-heart connections. (From which cerebral does areas preferentially heart innervation stem? How is organized the innervation at cardiac level? Does regional variations and/or preferential sympathetic rather than parasympathetic connections exist?)

Author Response

Dear Reviewer,

thank you for your valuable comments that can increase the quality. As proposed we addressed stated issues.

  1. We addressed the problematics of heterogenous clinical manifestation (Tako-tsubo, stunning, AMI) in patients after stroke with characterized possible differences in presentation.
  2. As suggested, we extended pathophysiology to more details at cell level, characterizing sympathicus and parasympathicus functions and proposed areas of influence and characterized lateralization in fifth paragraph.

Sincerely,

Mihalovic M